# How the Immune System Responds to Allergy Immunotherapy

**DOI:** 10.3390/biomedicines10112825

**Published:** 2022-11-05

**Authors:** Irene Veneziani, Nadine Landolina, Biancamaria Ricci, Oliviero Rossi, Lorenzo Moretta, Enrico Maggi

**Affiliations:** 1Department of Immunology, Bambino Gesù Children’s Hospital, IRCCS, 00146 Rome, Italy; 2Immunoallergology Unit, University Hospital of Careggi, 50100 Florence, Italy

**Keywords:** allergy immunotherapy, immuno-modulation and regulation, innate and adaptive response, pathogenic mechanisms

## Abstract

IgE-mediated diseases represent a highly diversified and multifactorial group of disorders that can deeply impact the patients’ quality of life. Currently, allergy immunotherapy (AIT) still remains the gold standard for the management of such pathologies. In this review, we comprehensively examine and discuss how AIT can affect both the innate and the adaptive immune responses at different cell levels and propose timing-scheduled alterations induced by AIT by hypothesizing five sequential phases: after the desensitization of effector non-lymphoid cells and a transient increase of IgE (phase 1), high doses of allergen given by AIT stimulate the shift from type 2/type 3 towards type 1 response (phase 2), which is progressively potentiated by the increase of IFN-γ that promotes the chronic activation of APCs, progressively leading to the hyperexpression of Notch1L (Delta4) and the secretion of IL-12 and IL-27, which are essential to activate IL-10 gene in Th1 and ILC1 cells. As consequence, an expansion of circulating memory Th1/Tr1 cells and ILC-reg characterizes the third phase addressed to antagonize/balance the excess of type 1 response (phase 3). The progressive increase of IL-10 triggers a number of regulatory circuits sustained by innate and adaptive immune cells and favoring T-cell tolerance (phase 4), which may also be maintained for a long period after AIT interruption (phase 5). Different administration approaches of AIT have shown a similar tailoring of the immune responses and can be monitored by timely, optimized biomarkers. The clinical failure of this treatment can occur, and many genetic/epigenetic polymorphisms/mutations involving several immunological mechanisms, such as the plasticity of immune responses and the induction/maintenance of regulatory circuits, have been described. The knowledge of how AIT can shape the immune system and its responses is a key tool to develop novel AIT strategies including the engineering of allergen or their epitopes. We now have the potential to understand the precise causes of AIT failure and to establish the best biomarkers of AIT efficacy in each phase of the treatment.

## 1. Introduction

Since its development in 1911, allergen-specific immunotherapy (AIT) still remains the only standard of care for the treatment of IgE-mediated allergic diseases, such as allergic rhinitis (AR), bronchial asthma (BA), and hymenopterum venom anaphylaxis (HVA). The goal of AIT is to ameliorate symptoms by inducing a stable immune tolerance to specific allergens, achieving long-term remission, preventing the onset of new sensitizations, and reducing progression of AR to BA [1,2,3,4,5]. The success of AIT relies on several factors, such as the selection of patients, allergen sensitization(s), administered doses and timing, product quality, and type and degree of the immune response. Currently, there are two main antigen-specific immunomodulatory treatments: subcutaneous immunotherapy (SCIT) and sublingual immunotherapy (SLIT), whereas several novel AIT approaches are being evaluated in clinical trials [6]. SCIT is an active tool for managing HVA, AR, and BA, while SLIT can be a valid alternative for those patients who develop local and, less frequently, systemic adverse reactions to SCIT [7]. A study including over 300 children with AR comparing the efficacy, safety, and compliance of SCIT and SLIT showed that the former is usually more effective than SLIT, whereas SLIT displays less-adverse reactions [8]. Other studies, however, demonstrated that SCIT in children is safe with a very low rate of severe side effects [9,10], but the risk of systemic reactions is greater in subjects with uncontrolled asthma and with accelerated dosing schedule [11].

Other alternative approaches to AIT have been described in the last decade. Oral immunotherapy (OIT) has been initially tried for respiratory allergens but was found not effective [12] except for OIT with encapsulated allergen extracts [13]. OIT is not effective for allergens easily digested in the gastrointestinal tract, such as the majority of respiratory allergens; thus, it is used in children mainly for food allergens containing digestion-resistant allergens such as peanuts, milk, and egg. The administration of 2 mg/d peanut protein for up to 5 years leads to a substantial and clinically meaningful desensitization coupled with good compliance and dosing safety [14].

Another approach is the intra-lymphatic immunotherapy (ILIT). As extensively reviewed [15], the basic strategy of ILIT is that the lymph nodes (LNs) are the best site to induce a rapid and strong immune response; thus, the direct exposure of allergen can promote a faster and more protective blocking of IgG antibodies (Abs) and immunomodulation than SCIT [16]. The major disadvantage of ILIT lies in the ultrasound guidance necessary to deliver the allergen into the LNs, making it a quite invasive procedure. 

As compared to other administration routes, the studies with epi-cutaneous immunotherapy (EPIT) are totally insufficient [17]. EPIT is a needle-free treatment, and hence, it was considered to be suitable for children. A recent review summarizing the results of trials [18] concluded that EPIT treatment can induce desensitization towards peanuts, associated with an increased risk of local adverse events (AEs). It uses high doses of the allergen and, although showing some improvement in seasonal symptoms, does not demonstrate significant benefits in terms of local side effects when compared with SCIT [19]. The table below summarizes the application of AIT mentioned above and their local or systemic side effects (Table 1).

This review aims to provide an update on the alterations of innate and adaptive immunity induced by AIT, to give an integrated view of deviation and regulation of immune responses differently operating along the treatment, and lastly, to analyze on the basis of the immunological response to AIT the causes of failure, the biomarkers of efficacy, and the new strategies to optimize therapy. 

## 2. Pathogenetic Mechanisms of Allergic Response

Allergic disorders, in particular BA, are multifactorial diseases in which genetic and environmental factors interact with each other [20,21]. BA displays a marked heterogeneity regarding etiology, symptom triggers and onset, clinical features, and response to therapy [22,23,24,25]. Indeed, the term asthma refers to a generic diagnosis including several clinical conditions (phenotypes) associated with distinctive functional or pathophysiologic mechanisms (endotypes) [26,27]. BA and other allergic diseases share endotypes or sub-endotypes [28] characterized by innate and adaptive immune responses modulated by non-allergic mechanisms (such as environmental factors, activated resident cells, or dysfunctional epithelial barrier) [29]. 

Genetic-epigenetic alterations influence the development of allergic diseases: a number of genetic loci have been associated with BA susceptibility, including the ORMDL3/GSDMB genes (childhood-onset asthma); deletion in the promoter region of the VEGF gene at position –2549 –2567, del 18 (irreversible bronchoconstriction) [30]; loci near the IL1RL1, IL33 (allergic asthma, present in distinct ethnic groups) [31]; TSLP (protective against T2 asthma as reviewed in [32]); PYHIN1 (present in individuals of African origins) [31]; and loci near IL13, RAD50, and IL4 genes (reviewed in [33]). Other genes involved in the development of BA have been fully reviewed [32].

According to the axiom that “genetics loads the gun and epigenetics pulls the trigger” [34], a number of epigenetic changes affecting gene expression and influenced by both environmental/social factors has been brought to attention [33,35]. Among these, DNA methylation, covalent post-translational histone modifications, and microRNA expression have been exhaustively reported [34,35,36,37,38,39].

In the frame of its genetic architecture, BA as well as other allergic diseases is mainly associated to increased levels of serum IgE in atopic individuals and allergic sensitizations [40,41], with inhalant allergens such as house dust mite (HDM), animal dander, cockroach, and seasonal pollens being the main triggers for its development [42].

The allergic inflammation is a two-step process with peculiar characteristics shared by all allergic diseases (reviewed in [43]). 

In allergic people, the priming with the allergen processed by antigen-presenting cells (APC) induces the expansion of allergen-specific CD4+ Th2 cells with the production of related cytokines (interleukin -IL-4, IL-5, IL-9, IL-13, and IL-31) that are responsible for the initiation, maintenance, and amplification of inflammation. The switch from naïve T cells to Th2 phenotype requires both IL-4 cytokine and cell-to-cell contact through Delta4 on DC and Notch1 on T cells [44,45]. Alarmins, such as IL-33 and IL-25, activate STAT6, cMaf, and GATA3 transcription factors, which promote chemokine receptor expression on Th2 cells for tissue homing [46]. Once activated, cells with type 2 profile are detectable in blood, tissue (as T resident memory, Trm2), and LNs (as T follicular helper cells, Tfh2). Notably, circulating Th2 and tissue Trm2 cells display distinct nonredundant function and distinct transcriptional profiles [47,48]. At the follicular level, Tfh2 and B cells’ cooperation, promoted by IL-4 and/or IL-13 with the interaction of CD154-CD40L expressed on activated T cells, induces the IgE switch and the expansion of allergen-specific IgE production by plasmablasts [48,49]. The binding between IgE and FcεRI, expressed by mast cells (MC) and basophils, persists after the first contact with the allergen. During a subsequent exposure, the allergen binds IgE fixed on MC and basophils, and their cross-link leads to the activation of PKC, PLA2, and MAP-kinases pathways with a consequent release of preformed (histamine, kinins, serotonin, etc.) and newly synthesized (eicosanoids such as leukotrienes; prostaglandins, PG; thromboxanes) mediators and several cytokines/chemokines responsible for immediate reactions and recruitment of multiple cell types [49]. These products have a crucial role in sustaining the immediate symptoms and late-phase events in which mobilized and activated neutrophils, monocytes, eosinophils, and lymphocytes lead to further secretory processes (reviewed in [43]). Endothelial cells upregulate the adhesion molecules, which favor the recruitment of blood cells promoted by chemokines, cytokines, and mediators such as histamine and leukotrienes released from MC, T, and other resident cells. Notably, eosinophil homing is favored by IL-5, IL-3, GM-CSF, and platelet activating factor (PAF) [50,51]. This late phase is characterized by the expansion of long-lived memory Th2- and mainly of Th2A cells, a Th2 subset specifically recognizing allergens and exclusively detectable in allergic individuals [52,53]. They are pathogenic and terminally differentiated T cells that express CRTH2, CD49d, CD161, and alarmin receptors; produce high levels of IL-5 and IL-9 [54]; and display distinct pathway of transcriptome. Importantly, circulating Th2A cells disappear in the favorable response to SCIT or OIT [55,56]. Th2 cells induce high production of IL-4 and IL-13 acting as IgE-switching factors, while IL-13 promotes production of inflammatory cytokines, goblet cell hyperplasia, airway hyper-responsiveness (AHR), and fibrosis. IL-5 is the major recruitment/expansion/survival factor for eosinophils, whereas IL-4, IL-9, and IL-13 favor MC homing. IL-9, IL-13, and IL-31 further damage the bronchi, worsening AHR. The chronic inflammation leads to airways remodeling characterized by subepithelial collagen deposition and increased angiogenesis, goblet and smooth muscle cell hyperplasia, and widespread epithelial damage [57,58].

Other immune cells contributing to develop, amplify, and maintain the allergic cascade are the type 2 innate lymphoid cells (ILC2). ILCs are tissue resident effector cells involved in lymphoid tissue formation, tissue remodeling after damage, and protection from pathogens [59]. Helper ILCs are the innate T helper counterpart and are divided into four groups (ILC1, ILC2, ILC3, and lymphoid tissue-inducer cells, LTi). ILC2 and ILC3 are involved in AR and BA pathogenesis, with ILC2 favoring eosinophil and ILC3 the neutrophilic prevalence [60,61]. ILC2 are detectable in tissues of allergic patients [62,63]. Alarmins play a pivotal role in ILC2 activation, starting allergic inflammation and favoring ILC2-Th2 cross-talk [61,64]. Activated ILC2 express GATA3/cMaf and produce IL-5, IL-13, low IL-4, and amphiregulin, favoring fibroblasts activation and airway remodeling [65]. Human-activated ILC2 expresses CD154, which potentially favors polyclonal IgE production during interaction with B cells [66]. The innate cellular response besides ILC2 includes eosinophils, basophils, MC, and macrophages. The Th2 cells plus ILC2, however, cannot fully explain the whole features observed in allergic inflammation, and other T-cell subsets have been shown to play an essential role in this process.

Th17 cells have been found as the new major actors involved in some asthma phenotypes usually associated with neutrophilic inflammation [67]. Their development from naive T cells is induced in humans by IL-1β and IL-23, which are highly produced by monocytes/macrophages following inflammasome activation [68]. Th17 cells promote immunity against fungal or extracellular pathogens [69,70] by producing IL-17A/F, IL-22, IL-6, and TNF-α. The release of proinflammatory cytokines and chemokines as CXCL8 due to IL-17A/F receptors engagement promotes neutrophil homing and increases in situ granulopoiesis. Th17 cells are also recruited into the skin or bronchial mucosa through CCL17 and CCL20 receptors, namely CCR4 and CCR6, respectively. High levels of IL-17A are found in the lung, induced sputum, bronco-alveolar lavage fluid (BALF), and serum of asthmatic patients upon allergen challenge. The increase in IL-17A levels observed in BALF was found to correlate with unspecific bronco-reactivity severity and obstruction, whereas some polymorphisms of IL-17F are associated with protection from BA [71]. ILC3 are highly producers of type 3 cytokines (IL-17A, IL-22, and GM-CSF) and thus might potentiate, at least to some extent, Th17-mediated inflammation [72]. This is in line with IL-17A-producing ILC3 cells, which were reported to be markedly elevated in the sputum and BALF of severe asthma [73,74] and ILC3 gene signatures that were enriched in cellular RNA from patients with adult-onset non-eosinophilic asthma. The type 3 immune response involves Th17 cells, ILC3, neutrophils, and macrophages.

Some subsets of unconventional T cells, such as γδT cells, specific invariant-natural killer T (iNKT), and mucosal-associated invariant T (MAIT) cells, are involved in pathogenic aspects of neutrophilic inflammation, mainly producing IL-17A [75,76,77]. IL-17A can also be produced in the lung by alveolar macrophages and epithelium, favoring the neutrophil influx, the production of pro-fibrotic cytokines by fibroblasts, and the release of eosinophil chemo-attractants by the airway muscle cells [3].

Based on the prevalence of the two major types of effector T cells (Th2 and Th17) and helper ILCs (ILC2 and ILC3), endotypes of BA have been distinguished into “type 2” (or T2 high or eosinophilic) and “non-type 2” (or T2 low or non-eosinophilic) [24,78,79]. Mixed endotypes have also been described [80]. T2 high endotype is commonly induced by allergic inflammation sustained by type 2 immune response, characterized by the presence of tissue eosinophils [81]. In contrast, the T2 low endotype includes heterogeneous conditions unrelated to allergy and/or eosinophilic inflammation, as found in neutrophilic asthma and pauci-granulocytic asthma, which are prevalently sustained by type 3 immune response [82]. In conclusion, a successful AIT must shift both type 2 and type 3 immune responses into a more protective profile.

## 3. Alterations of Innate and Adaptive Immunity Induced by AIT

Initial studies provided evidence of a reduced sensitivity and reactivity of basophils and MC and their altered distribution in tissues associated to AIT [3,83]. However, after having shown increased blocking antibodies and reduced proliferation of memory T cells to allergens in treated patients, research on AIT-mediated alteration has been mainly focused on innate and adaptive immunity [84]. As reported, T-cell unresponsiveness to an allergen might be due to high-dose tolerance because doses given in traditional SCIT are higher than those encountered naturally [85,86]. The anergy of allergen-specific T cells was described in studies of SCIT for HVA, where the impaired T-cell response to phospholipase A2 (the major bee allergen) brought to a reduction in the proliferative response through IL-10 [87]. Additionally, high-dose antigen can also trigger apoptosis of Th2 cells in allergen-stimulated PBMC from treated patients [88]. Anergy and apoptosis, however, have been usually observed with prolonged AIT, whereas, likely, the early alterations are essentially due to functional modifications of innate (ILCs) and adaptive (T and B) cell profiles. 

### 3.1. The Response of B-Cell Compartment to AIT

The humoral response to AIT includes the early transient increase of IgE (which rapidly declines), followed by a sustained upregulation of blocking allergen-specific IgG1 Abs. Along the treatment, IgG1 is substituted by IgG4 Abs, with non-inflammatory properties, since they are unable to trigger complement cascade and cell activation [89], while they compete with effector cell-bound IgE, preventing activation of MC and basophils [4]. The balancing of IgE, IgG1, and IgG4 is mediated by a panel of cytokines, prevalently IFN-γ and IL-10 [90,91]. Increasing evidence underlines the importance of allergen-specific IgA Abs in AIT; they induce protection by competing with IgE for allergen recognition, thereby preventing IgE-mediated reactions and reducing inflammatory responses through FcαRI signaling and the induction of a DC-T regulatory response [92]. Secretory IgA offers a first line of protection by blocking mucosal allergen absorption; but, upon systemic access of the allergen, which is required for IgE-mediated anaphylaxis, allergen-specific IgA (and IgG) constitutes the last guard [92]. In infants, the development of asthma was associated with decreased proportion of IgA bound to fecal bacteria [93], whereas in 6-month infants with high fecal IgA levels, the risk to develop allergic diseases within 2 years tended to reduce significantly [94]. Mucosal and serum allergen-specific IgA have been found to increase in OIT for peanuts [95] and in AIT for pollens [96]. Notably, in these latter patients, SLIT induced higher increase of IgA1 and IgA2 in nasal fluids and of IgA1 in the serum compared to SCIT [97]. 

Importantly, a heterogeneous subset of regulatory B (B reg) cells with anti-inflammatory function has been described in patients undergoing AIT. Growing evidence indicates that B reg cells play a role in inducing and maintaining allergen tolerance [98,99]. Along AIT, B-reg-cell-derived plasma-blasts produce blocking allergen-specific IgG (especially IgG4) Abs due to their high production of IL-10 (the best switch factor for IgG4 subclass) during AIT. Accordingly, after two years of AIT for pollens, IL-10+B cells increased both in mucosa and periphery, strictly correlating with the levels of allergen-specific IgG4. Importantly, patients’ sera showed blocking IgG activity correlated with their overall improvement [100]. In agreement, healthy beekeepers and patients undergoing AIT for HVA showed IL-10+B cells highly producing IgG4 [101]. This B-cells subset, named Br1, is characterized by CD25+CD71+CD73- phenotype and potently inhibits T-cell response to allergens in an IL-10-dependent manner [102]. 

Besides IL-10, other cytokines such as TGF-β and IL-35 as well as some surface-bound suppressive molecules have also been shown to contribute to B-reg-cell-suppressive activity. In a double-blinded, placebo-controlled food challenge (DBPLFC) study, it was found that the proportion of TGF-β-producing CD5+B reg cells was higher in the milk-tolerant case series than in the milk-allergic group [103]. TGF-β+ B reg cells can promote T regulatory (T reg) cell development and apoptosis of T effector cells, thus contributing to regulate allergic inflammation. 

### 3.2. AIT-Induced Immunomodulation of T Cell Response

The induction of specific IgG and IgA Abs by AIT clearly indicates the prominent role for memory T effector cells: producing switching factors for IgG1/4 and IgA1/2 subclasses. On the other hand, the impairment of T-cell response to allergens and reduced activity of MC and basophils also suggest a relevant role for T reg cells.

#### 3.2.1. Immune-Deviation

Initial studies from the 1990s based on several in vitro and in vivo models concluded that AIT skewed allergen-specific responses from Th2 to a more protective Th1 profile, a mechanism defined as immune deviation [104]. As firstly documented by our group [105] and further confirmed by others [106,107], fully polarized Th2 or Th17 (as ILC2 and ILC3) cells cannot be considered terminally differentiated but rather plastic entities. For instance, IL-12 can epigenetically modulate both Th2 and Th17 responses towards a Th1 direction even though Th1 progeny can maintain some features of the original cells [108,109]. Th17-derived Th1 cells have been defined as “non-classical Th1” or “Th1 stars”, which were found in BALF of children with severe BA [110]. Similar mechanisms are also found in other tissue-resident cells such as Trm-, Tfh-, γδT, NKT, MAIT, and T reg cells [111,112]. ILCs are highly flexible upon environmental signals, likely due to the abundance of cytokine receptors able to promote shifting effects from one to another ILC subset [113,114].

The shift towards Th1- has been associated to different AITs [115,116], as demonstrated for the T2 high endotype upon treatment with biologicals (as anti-IL-4R monoclonal antibodies) [117]. In AIT-treated patients, Th1 cells were increased in the nasal mucosa after pollen challenge [118], and they inversely correlate with symptom and medication scores. Additionally, a similar pattern was described for the effector cytokines. Indeed, the increase of IFN-γ and the parallel decrease of IL-5 and IL-9 during the pollen season were shown in nasal biopsies and fluids of AIT-treated patients [119,120]. SCIT usually inhibits allergen-induced late cutaneous responses, and this correlates with increased IL-12 expression in skin macrophages [121]. The dual increase of IL-12 by monocytes and IFN-γ by NK and T cells was also reported [122]. In conclusion, AIT-mediated immune-deviation is widely detectable at tissue and systemic levels [3,123].

#### 3.2.2. Immune-Regulation

The importance of increased activity of T reg cells (immune regulation) as the main mechanism to explain the efficacy of AIT was emphasized more recently. Adaptive T reg cells originate in LNs during the priming of T effector cells, and their activity is mediated by both regulatory cytokines (IL-10, TGF-β, IL-35, and IL-37) and cell-to-cell contact mechanisms [124]. During pollen exposure, T reg cells from grass-sensitive patients are impaired in inhibiting type 2 cytokines compared to those from healthy donors (HD) [125]. However, our group showed that Dermatophagoides pteronyssinus 1 (Der p 1)-specific FoxP3+T reg cells expanded from the PBMC from HD exert similar phenotype and function of those derived from allergic donors. Moreover, increased proportions of T reg cells producing IL-10 (Tr1) have been described during AIT treatment, indicating some in vivo role for these cells. IL-10 inhibits major histocompatibility complex (MHC) class II expression on DC or the tyrosine phosphorylation of CD28 in T cells, preventing downstream signaling events [126]. Furthermore, IL-10 downregulates the type 1/2/3 cytokines in vitro, inducing anergy of T effector cells [127]. The regulatory cytokines were increased mainly in the blood and skin of patients treated with AIT for HVA [87]. Tr1 cells have been also detected in nasal mucosa in AR patients [128]. Increased local Tr1 cells have been associated with elevated IgG4 and blocking activity of IgE-facilitated allergen presentation [129]. During AIT, increased T reg cells have been also related to high production of TGF-β, which, with that derived from B reg cells, regulates antigen presentation, co-stimulatory molecule expression, T-cell proliferation, and IgA switch [130].

Recent findings highlight that other regulatory factors produced by T and other inflammatory cells play a key role in the efficacy of AIT, in particular IL-35 and IL-37. There are no current studies on the activity of IL-37 during AIT, but due to what is known so far, it is reasonable that this new cytokine may play a role in allergen-specific induced tolerance by reducing IgE, eosinophils, and type 2 response [131,132]. By contrast, data on IL-35 are more promising. IL-35 belongs to the IL-12 family and is produced by Foxp3-T reg, B reg, endothelial and smooth muscle cells, as well as monocytes. rIL-35 suppresses T-cell proliferation and Th17 differentiation, expands Foxp3+T reg cells [132,133], or induces a subset of Foxp3-negative inducible T reg35 cells (iTr35) exclusively producing IL-35 [134,135]. In mice, IL-35 is selectively produced by ICOS+T reg cells and reversibly inhibits allergic inflammation sustained by type 2 and type 3 responses [136]. In agreement, it was found that, in pollen AIT, IL-35 produced by iTr35 cells strongly inhibits type 2 immune responses, IgE production from B cells, DC priming of Th2 responses, and converted Th2 cells into immunosuppressive iTr35 cells [137]. Notably, SLIT restored iTr35 cells in treated patients. These findings provide a rationale for IL-35 therapy for the treatment of respiratory allergic diseases.

### 3.3. AIT-Induced Modulation of Innate Lymphoid Cells

Several reports support the hypothesis that AIT can modulate ILC populations. Upon AIT for HDM allergy, responder patients and HD showed a decreased frequency of circulating ILC2 compared to non-responder patients. Conversely, ILC1 from responders and HD were increased compared to non-responder patients. Moreover, PBMC from responder patients had a significantly lower expression of activation markers on ILC2 upon allergen re-stimulation compared to non-responders [138]. The mechanism of ILC2 in modulating symptoms during AIT has been partially clarified with the discovery of two functionally distinct ILC subsets. KLRG1+ but not KLRG1– ILC2, produced IL-10 upon activation with IL-33 and retinoic acid and are also named ILC-reg. These cells reduced Th2 responses and maintained epithelial cell integrity. The proportion of IL-10+KLRG1+ILC2 was lower in patients with grass-pollen allergy when compared to HD, and the ability to produce IL-10 by ILC2 was restored in patients receiving AIT [139]. This highlights the relevance of IL-10 in AIT-mediated tolerance and as a biomarker for a successful AIT.

## 4. AIT-Induced Immune Deviation and Immune Regulation Are Two Related and Sequential Phases of the Chronic Stimulation with Allergen

In the last two decades, the mechanisms of immune deviation and immune regulation induced by AIT have usually been presented as alternative, mutually exclusive events, and often, the discussion of these two paradigms has deteriorated into a permanent, unnerving, and endless debate. More recent data, however, indicate that, likely, the supposed dualism actually may be considered as two sequential immune responses due to the chronic stimulation with allergen. The ability of an antigen to induce both IL-10 and IFN-γ was described for the hepatitis C virus core protein as a general homeostatic mechanism favoring the persistence of infection [140].

Our group showed that Par j1 allergen sequence contains epitopes with different immunoreactive (Th1 or Treg) potential [141]. In agreement, AIT using Fel d1 peptides contains two major epitopes inducing IL-10 or IFN-γ [142], and the magnitude of the cutaneous reaction paralleled the accumulation of Tr1 and Th1 cells in allergen-challenged skin sites [143]. A connection between IFN-γ and IL-10 production was described initially in experimental models in which CXCL10 (a type-1-related chemokine) could shift mature- into a tolerogenic DC able to expand T reg cells in vitro [144]. On the other hand, the chronic stimulation of DC by IFN-γ could induce the hyperproduction of IDO, whose metabolites block T-cell proliferation and expand Tr1 cells [145,146]. In ultra-rush AIT for HVA, the shift from Th2 to Th1 cell profiles has been described, with increase of T reg cells producing or not IL-10 [146].

Some authors suggested that the predominance of some mechanisms of tolerance is time- and dose-dependent. SLIT has been shown to induce IL-10-producing cells in the early (1 month) step of treatment, followed by an IFN-γ (plus or not IL-10) skewing after 1 year of therapy [147]. In another study, the effective SLIT was associated with the early increase of TGF-β and the late up-regulation of IFN-γ and IL-10, both produced by T cells [148]. Lastly, the increase of IL-10+T reg cells in the nasal mucosa of AIT- or in the blood of OIT-treated patients was associated with a parallel upregulation of memory Th1 cells [128,149].

The turning point came with the report of our group on immunological response to HDM in AIT-treated patients, where we showed that, after 6 months of therapy, circulating memory T cells co-expressed high amounts of both IL-10 and IFN-γ, a feature not detectable before treatment [150]. Importantly, beyond AIT, the contemporary synthesis of IL-10 and IFN-γ from the same T cell has been reported in pathological conditions including several infections, autoimmune disorders, and cancer, in which a chronic antigen stimulation occurs [127,151,152,153,154,155,156,157,158].

Indeed, even though the IL-10 promoter is silenced in Th1 cells, the IL-10 locus is in a reversible histone deacetylase-responsive state, which can be re-activated when a prolonged antigenic stimulus, a high activation state, and elevated IL-12 levels are present [159,160]. The Th1/Tr1 cells are probably induced to impair the collateral damage caused by high levels of inflammation even though they reduce the efficacy of the immune response. In the presence of IL-12 and IL-27 [160], Notch1 engagement is the main signal to induce IL-10 production in already-established Th1 cells via a signal transduction and activator of transcription-4 (STAT4)-dependent pathway. 

Both IL-10 and IFN-γ are able to impair in vitro allergen-specific Th2/Th17 responses, down-regulating IgE-producing cells (via IFN-γ) or switching to IgG4 (via IL-10), highlighting that immune deviation and immune regulation are two faces of the same coin.

This gives a rationale to explain the clinical efficacy of AIT. Th1/Tr1 cells have been described in mice upon chronic stimulation with antigens and are involved in humans in the protection against pathogens such as leishmania, borrelia, and mycobacteria [160]. Of note, many reports indicated that the main source of IL-10 in chronic human infections was Foxp3-Th1 cells [161,162].

Based on the previous in vivo and in vitro findings, a tentative timing schedule of alterations induced by AIT can thus be proposed (Figure 1). Five sequential time phases can be hypothesized. First, we can observe the desensitization of effector non-lymphoid cells of allergic cascade, such as MC and basophils, which has been described as the most precocious event. Indeed, it has been documented that a high dose of allergen administered with AIT may induce a transient small increase of IgE (activation of tissue ILC2, Trm2 and lymph nodal Tfh2 cells) associated with activation of effector cells (MC and basophils) by allergen-IgE immunocomplexes. This is followed by the downregulation of FcεRI, histamine receptors, and cytokines promoting type 2 response. The excess of allergen locally stimulates phagocytic cells, which produce cytokines (such as IL-12 and IL-18) essential to orient T and ILC towards a type 1 profile (Th1, Tc1, and ILC1). This second phase (shift to type 1 response) is progressively potentiated by the increase of IFN-γ, which, in turn, further promotes APC to produce cytokines favoring type 1 response and IDO, further blocking effector cells. The ongoing allergen stimulation promotes the chronic activation of macrophages/monocytes/DC by IFN-γ progressively leading to the hyperexpression of Notch1L (Delta4) plus the secretion of IL-12 and IL-27, which are the signals essential to activate IL-10 gene into Th1 and ILC1 cells. This represents a physiologic response linked to chronic antigen stimulation, leading to expansion of circulating memory Th1/Tr1 cells and ILC-reg; it characterizes the third phase addressed to antagonize/balance the excess of type 1 response. The progressive increase of IL-10 conditions many cells of adaptive (T reg, B reg, Tr1, and Tfh-reg) and innate (ILC-reg, DC-reg, etc.) immunity to assume a regulatory profile (fourth phase characterized by the regulatory response). In turn, these cells increase their suppressive potential with the production of other regulatory cytokines (such as IL-35 and TGF-β), which further orient APC (prevalently the DC reg) towards the full tolerance of allergen response with the secretion of high IL-10 and IL-35 levels (five phase of amplified full regulation) (Figure 1).

This sequence of events has been also observed during the chronic administration of biologicals (which must be considered exogenous antigens) for several immune-mediated diseases. Indeed, in a longitudinal study, we found that after 8 months of treatment, IL-10 is stably expressed by infliximab-stimulated PBMC in tolerant patients, which downregulates adaptive cytokines (IFN-γ) induced in the early phase [163]. Similarly, the rapid desensitization procedure for biologicals in patients with previous anaphylaxis due to the anti-drug IgE Abs also mimics the AIT mechanisms. In this case, the early reduction of IgE-anti-drug antibodies (ADA) and skin tests for biologicals is associated with the early increase of IgG ADA, which declines or disappears later on. In parallel, the early increase of drug-specific T cells producing IFN-γ, which becomes negative at the end of treatment, correlates with the late increase of memory Tr1 and Tr35 cells in vitro and the increase of circulating regulatory cytokines [164,165,166].

## 5. Causes of Failure, Biomarkers of Efficacy, and New Strategies to Optimize AIT

### 5.1. Causes of AIT Failure

Based on the previous sequence of AIT-related immune alterations leading to the full allergen tolerance, it is possible to hypothesize the potential causes of AIT failures.

The first and likely most relevant causes of AIT failure are the genetic or epigenetic polymorphisms of cells (Th2/Th17 and ILC2/ILC3) and molecules (cytokines and receptors) inducing the development and the effector phase of types 2/3 immune responses of allergic inflammation. Alterations of cytokines (such as IL-12, IL-18, and IL-27) and molecules able to shift these responses as well as their receptors and their signaling pathways could be cause of unsuccessful AIT. Moreover, another possibility of failure is the onset of dysfunctional Th1/Tr1 cells, likely due to the polymorphisms of Notch ligand expression and molecules involved in their signaling. Another important cause of AIT failure can be due to impaired IFN-γ/IFN-γR interaction and molecules involved in the signaling pathway. Genetic alterations of receptors (as such H2R, FcεRI, etc.) or secreted molecules (IDO, etc.) may further contribute to AIT failure. Finally, polymorphisms of regulatory cytokines (IL-10, IL-35, IL-37, and TGF-β), their receptors and signaling pathways may be one of the main factors responsible of unsuccessful AIT. The correlation between some previously described polymorphisms and AIT failure can be documented by using big data deriving from international AIT registries.

The recent SARS-CoV2 pandemic and the need to administer the mRNA-based or viral peptide-based vaccines pose the question of a potential incompatibility with AIT. Based on well-known immunological mechanisms, COVID-19 vaccines are addressed to induce a valid humoral and cellular adaptive response through few doses of the reagent, while AIT is aimed to enhance a specific immune tolerance through several repeated doses of allergens. Both treatments are totally specific, thus involving different types of T and B cells not interfering each other. Based on this knowledge, the use of the four approved COVID-19 vaccines does not represent a contraindication for AIT. However, to identify possible reactions to one of the two vaccines, it is advisable to space the two administrations by at least a week both in the case of a starting AIT or a treatment in progress [167].

### 5.2. Biomarkers of Efficacy

According to European Academy of Allergy and Clinical Immunology (EAACI), potential biomarkers for monitoring the clinical efficacy of AIT are divided into six domains: (a) antibodies (total IgE, allergen-specific IgE, IgG1, IgG4, IgA1, and IgA2), (b) serum inhibitory activity of IgE (IgE Fab and IgE BF), (c) basophil activation markers (phenotype of allergen-activated basophils: CD63, CD203c, etc.), (d) cytokines (IL-10, IL-13, IFN-γ, IL-18, IL-27, IL-35, IL-37, TGF-β, IL-4, and IL-5) and chemokines (CCL11/24/26 and CXCL9/10/11), € cellular biomarkers (Foxp3+Treg cells, Tr1, Tr35, TfhR, B reg cells, DC reg, ILC2-reg, Th1-ILC1, Th2-ILC2, Th17-ILC3, etc.), and (f) in vivo (the previous domains after allergen challenge or during chamber study) [168].

Taking into account the sequence of immunologic events along AIT treatment, the next challenge is to define the best timing for monitoring different domains from the beginning of AIT and the efficacy range of each biomarker.

During the first phase (desensitization of effector cells), the following parameters can be analyzed: increased serum IgE, low expression of FcεRI on MC and basophils, monocytic apoptosis, and slight increase of T-cell response to the allergen in vitro.

The second phase (shift to Th1 response) is characterized essentially by increased specific IgG1 and decline of IgE, upregulation of allergen-specific T cells and ILC exerting a type 1 profile, and the decrease of type 2 and 3 cytokines/chemokines in favor of IFN-γ or CXCL9/10/11. Importantly, the disappearance of circulating Th2A cells can be considered a suitable marker of favorable response to following SCIT for pollens or OIT for peanuts [55,56].

The third phase characterized by the allergen-specific Th1/Tr1 cells can be monitored by the reduction of allergen-specific IgG1 and increased IgG4/IgA1/IgA2, the presence of allergen-specific Th1/Tr1 cells and of IL-10+ILC2 (ILC2-reg), and the increased serum IL-10 upon allergen exposure. 

The last two periods (onset of the regulatory response and its amplification phase) can be monitored by the high proportions of regulatory subsets (Foxp3+T reg cells, Tr1, Tr35, TfhR, ILC reg, DC reg, and B reg cells); the increased levels of TGF-β, IL-10, IL-35, and IL-37; and the poor or no response of T cells to allergen in vitro. The presence of memory regulatory cells after treatment interruption can be a useful prognostic parameter of symptoms’ reduction.

### 5.3. Strategies to Optimize AIT

An effective pathogenic AIT should fully redirect the allergen-specific type 2/type 3 responses to a protective and balanced type 1/Tr1 phenotype, which amplifies many regulatory circuits. Currently, the relative low proportion of patients with good outcome, the potential side effects, and the long-term use, which may reduce the efficacy, are valid reasons to improve this treatment [169]. 

AIT with peptides has been studied for more than 20 years in clinical trials using cat [122], HDM [170], and pollen [171] peptides, but inconclusive results have been obtained [172]. Vaccines with long peptides (20–40aa) prevalently bind IgE, whereas vaccines with short peptides (10–17aa) impact with T-cell response since are recognized by MHC Class II molecules on DC [173,174]. In this case Th2/Th17 cell response shift towards a Th1/Tr1 profile, where IL-10 decreases eosinophils, basophils, and MCs recruitment to tissues [175]. 

Allergen derivatives from recombinant allergens can be rendered hypo-allergenic by genetic engineering or chemical modification [176]. The efficacy of recombinant allergens is equivalent or even more favorable than native allergen extracts [90,177]. The development of recombinant hypo-allergens is represented by carrier-bound B-cell epitope-containing peptides [178].

Virus-like particles (VLPs) are another possible basis for the development of AIT [179]. VLP-based vaccines are well-tolerated and immunologically effective towards papillomavirus, hepatitis B virus, and malaria [180]. VLPs have a diameter in the range of 30 nm, a crucial size to be drained into the lymph nodes, where an optimal B-cells activation occurs. The VLPs surface can be modified to display the allergen epitopes, while the inside of VLPs can be loaded with some TLR ligands, such as RNA or DNA sequences activating TLR7/8 and TLR9, respectively. When VLPs are inside the APC, the antigen is processed and its peptides presented in association with MHC Class I and Class II molecules. Some phase I clinical trials designed to assess the impact of VLPs plus TLR ligands on AR have been performed to orient the response to allergen towards a Th1/Tr1 profile. In all, a decreased production of IgE with a parallel increase of IgG4 specific for the allergens has been observed [181,182,183,184].

Finally, another strategy to improve a Th1 response to allergens is to bind allergen (or its peptides) with endosomal TLR ligands. In the last few years, our group demonstrated how OVA or nDerp-2 allergens coupled with a modified adenine (a TLR7 ligand), can switch allergen specific T-cell response from a Th2/Th17 to a Th1 profile in vitro and in vivo. In this model, a decreased production of IgE and a parallel increase of IgG2a and IFN-γ were found [185,186].

## 6. Conclusions

The recent findings on the mechanisms of AIT clearly indicate that this chronic allergen stimulation primarily affects the tissue and/or lymph nodal APC, modifying environmental milieu of allergic inflammation, which, in turn, modulates innate and adaptive immune responses at different cell levels.

In addition, AIT induces the early switch from type 2/type 3 to type 1 response (immune deviation), while the chronic ongoing therapy favors the IL-10 production by memory Th1 cells. This is the initial mechanism triggering a number of regulatory circuits and favoring the loss of response to allergen (immune regulation), which may be maintained for a long period after AIT interruption. 

Importantly, all the known AIT procedures, such as SCIT, SLIT, OIT, ILIT, and EPIT, have been shown to induce similar immunological changes and can be monitored by timely, optimized biomarkers.

AIT failure may be due genetic/epigenetic polymorphisms of several mechanisms, including the plasticity of immune responses and the induction/maintenance of several regulatory mechanisms sustained by cells and cytokines. Finally, novel AIT strategies include engineering of allergens or their epitopes, which must take into account the previous described alterations as the onset of Th1/Tr1 cells, which are the trigger of an efficient tolerance to an allergen. We now have the potential to understand the precise causes of AIT failure as well as to define the best biomarkers of AIT efficacy; it will be possible by using big data deriving from international AIT registries.

## Figures and Tables

**Figure 1 biomedicines-10-02825-f001:**
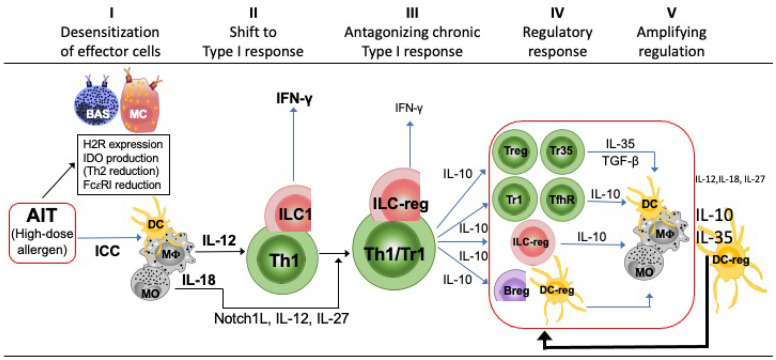
Timing schedule of alterations induced by AIT. Five sequential time phases can be hypothesized, named from I to V in the figure and described in the text.

**Table 1 biomedicines-10-02825-t001:** Types of immunotherapy and their application in allergic diseases.

	HVA	AR	BA	FA	LR	SR (Which Require Epinephrine Treatment)	Ref
SCIT	X	X	X		Erythema, pruritus, and swelling at the injection site	Low frequencies	[7,9,10]
SLIT		X	X		Oropharyngeal pruritus, swelling, or both; throat irritation; nausea/vomiting; diarrhea; abdominal discomfort; heartburn; and uvular edema	Uncommon	[8,9,10]
OIT				X	Oral pruritus, abdominal discomfort, and rashes	Common during home dosing	[14]
ILIT		X			Local swelling at the injection site	Uncommon	[14]
EPIT		X		X	Local eczematous reactions	Uncommon	[18]

SCIT, subcutaneous immunotherapy; SLIT, sublingual immunotherapy; OIT, oral immunotherapy; ILIT, intra-lymphatic immunotherapy; EPIT, epi-cutaneous immunotherapy; AR, allergic rhinitis; BA, bronchial asthma; HVA, hymenoptera venom anaphylaxis; FA, food allergy; LR, local reaction; SR, systemic reaction; X indicates that the AIT listed to the left are a feasible therapeutic option for each allergic diseases.

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
