# Peer review of "How the Immune System Responds to Allergy Immunotherapy"

_biomedicines, 2022, doi:10.3390/biomedicines10112825_

Round 1

Reviewer 1 Report

I think this is a detailed review of the mechanisms by which allergen immunotherapy appears to be effective. I believe that the results based on basic research on the mechanism of immunotherapy will be useful information for clinicians. It also includes the latest data, so you can learn about the global research situation. I hope to provide useful information to our readers.

Author Response

We thank the Reviewer for the positive comments on our work.

Reviewer 2 Report

The work submitted for evaluation by Irene Veneziani et al.How the immune system responds to Allergy Immunotherapy” makes accessible the immune mechanisms of immunotherapy.

This is a valuable review for practitioners and allergists who want to explore these issues.

The work is written in a good language, contains a lot of important information and is based on the results of recent research.

Reviewer's comments:

1. In the Introduction section - please add a table or a figure taking into account the types of immunotherapy (SCIL, SLIT etc.) with a brief description of the application - then the work will gain in value.

2. In the section "Pathogenetic machanisms of allergic response" it is worth quoting the article "The -2549 -2567 del18 polymorphism in VEGF and irreversible bronchoconstriction in asthmatics" J. Investig.Allergol.Clin.Immunol. 2019 Vol. 29 No. 6 pp. 431-435.

3. In many places in the article (e.g. lines 231, 285, 298, 318, 378, 402, 416, 471, 555) the letters/marks after i.e., IFN, TGF, Fc RI appear to be missing.

4. In line 258 please add the abbreviation DBPLFC after ”double-blinded placebo-controlled food challenge”.

5. The article is written as Review, so what is the role of the paragraphs: line 263-266, 351-354, 435-443?

Best regards.

Author Response

We thank the Reviewer for the useful comments. The following improvements have been made in the new version of our manuscript:

  1. The table was added at page 3;
  2. The article "The -2549 -2567 del18 polymorphism in VEGF and irreversible bronchoconstriction in asthmatics" J. Investig.Allergol.Clin.Immunol. 2019 Vol. 29 No. 6 pp. 431-435, was added as Ref. 30;
  3. Greek letters were corrected;
  4. The suggested abbreviation was added to the text;
  5. The paragraphs not related to the review were deleted.

Reviewer 3 Report

The review by Veneziani et al. gives a comprehensive overview of what is known about the immune mechanisms that are at play in allergen-specific immunotherapy. It tackles the question what could be a reason for the failure of AIT and mentions the novel ideas that have been developed to improve AIT.

The paper is a pleasure to read, it gives a very informative summary of the current knowledge about allergic disease mechanisms and the way in which AIT interferes with them and improves the disease. The division of the immune response into 5 different phases makes sense and so does the idea of having to use diverse biomarkers to control for the success of AIT at the different stages. Especially, since a universal biomarker has so far been elusive.

I have only some minor remarks to make:

Having read the excellent paper and then re-reading the abstract gave me the feeling that the abstract does not give full credit to the paper. I would suggest some revision to emphasise the important findings better and to reflect the structure of the paper more accurately.

Line 170: “…observed in allergic inflammation and other cells have been shown to play…”: this sentence would make more sense to me if it read: “…observed in allergic inflammation and other T cell subsets have been shown to play…”

Line 214: “…on innate and adaptive immunity…”: should this not read “on adaptive immunity”?

There are three instances where information from a different paper seems to have slipped into the manuscript: Lines 262 – 266, 350 – 354, 434 – 442

In some instances in my version of the paper Greek letters are not visible, f.ex. in FcεRI or INF-γ.

Author Response

We thank the Reviewer for the useful comments. The following improvements have been made in the new version of the manuscript:

  1. We modified the abstract emphasizing the proposed model (novelty of our review);
  2. Line 170 (current Line 164) has been properly modified;
  3. Line 214 (current Line 201) not modified because ILCs are described as well and represent the innate counterpart;
  4. The paragraphs not related to the review were deleted;
  5. Greek letters were corrected
  6. The highlighted paragraphs in the previous version of the manuscript  have been changed and rephrased in the revised version.

Round 2

Reviewer 2 Report

Dear Authors,

all my the comments have been included in the revised version of the article. Good luck with your publication and further work.

Regards.